# Potential Biological Activities of Peptides Generated during Casein Proteolysis by Curly Kale (*Brassica oleracea* L. var. *sabellica* L.) Leaf Extract: An In Silico Preliminary Study

**DOI:** 10.3390/foods10112877

**Published:** 2021-11-21

**Authors:** Magdalena Polak-Berecka, Magdalena Michalak-Tomczyk, Katarzyna Skrzypczak, Katarzyna Michalak, Kamila Rachwał, Adam Waśko

**Affiliations:** 1Department of Biotechnology, Microbiology and Human Nutrition, University of Life Sciences in Lublin, Skromna 8, 20-704 Lublin, Poland; magdalena.michalak@up.lublin.pl (M.M.-T.); kamila.rachwal@up.lublin.pl (K.R.); adam.wasko@up.lublin.pl (A.W.); 2Department of Animal Physiology and Toxicology, Faculty of Science and Health, The John Paul II Catholic University of Lublin, Konstantynów 1H, 20-708 Lublin, Poland; 3Department of Plant Technology and Gastronomy, University of Life Sciences in Lublin, Skromna 8, 20-704 Lublin, Poland; katarzyna.skrzypczak@up.lublin.pl; 4Department of Epizootiology and Clinic of Infectious Diseases, Faculty of Veterinary Medicine, University of Life Sciences in Lublin, Głęboka 30, 20-612 Lublin, Poland; kat.michalak86@gmail.com

**Keywords:** bioactive peptides, casein, curly kale, MALDI–TOF

## Abstract

This study is a brief report on the proteolytic activity of curly kale leaf extract against casein. Casein degradation products and an in silico analysis of the biological activity of the peptides obtained was performed. The efficiency of casein hydrolysis by curly kale extract was determined using SDS–PAGE and by peptide concentration determination. The pattern of the enzymatic activity was determined by MALDI–TOF MS analysis. The results showed that α- and β-casein were more resistant to curly kale extract hydrolysis, whereas κ-casein was absent in the protein profile after 8 h of proteolysis, and all casein fractions were completely hydrolyzed after 24 h of incubation. Based on sequence analysis, seven peptides were identified, with molecular mass in the range of 1151–3024 Da. All the peptides were products of β-casein hydrolysis. The identified amino acid sequences were analyzed in BIOPEP, MBPDB, and FeptideDB databases in order to detect the potential activities of the peptides. In silico analysis suggests that the β-casein-derived peptides possess sequences of peptides with ACE inhibitory, antioxidant, dipeptidyl peptidase IV inhibitory, antithrombotic, immunomodulatory, and antiamnesic bioactivity. Our study was first to evaluate the possibility of applying curly kale leaf extract to generate biopeptides through β-casein hydrolysis.

## 1. Introduction

Proteases play a role in the food industry by improving the functional and nutritional properties of proteins [1]. In food processing, especially in the dairy industry, proteases play a significant role in milk coagulation. Increased global cheese production and a worldwide shortage of rennet have necessitated the search for new proteases with rennet-like properties [2]. These alternatives include plant proteases [3,4]. Plant proteases can also be used to manufacture cheese for lactovegetarian consumers and organic markets, as well as for kosher and halal products [5]. However, the excessive proteolytic nature of many so far studied plant coagulants has limited their use in cheese manufacturing, and there remains a need to find new potential milk-clotting enzymes from plants in order to meet the increasing global demand for diversified cheese of good quality [2]. The wide variety of plant proteases with unique catalytic properties, differentiated specificity, and stability, has made their future commercial application particularly attractive. Further research on the identification and characterization of these newly emerging plant proteases, in order to select an appropriate enzyme and the design of an efficient reasonable hydrolytic process, is needed [6,7]. A previous study [8] has revealed that broccoli shows protease activity, which was one of the most promising activities among 90 plant resources. Knowing the proteolytic possibilities of the *Brassica* genus, we examined curly kale juice and reported that it can be a potential and effective coagulant in the production of feta-type cheese. The use of kale extract as a coagulating agent in the food industry is safe, and due to its well-studied and described health-promoting properties, it increases the functional value of the product [9].

Casein hydrolysate is a source of bioactive peptides, which are not active in the parent protein but are released and activated by enzymatic hydrolysis (including as a consequence of proteolysis by enzymes derived from plants) [10]. Many studies have confirmed that casein-derived biologically active peptides behave as regulatory components in the body, with a hormone-like activity that may modulate specific physiological functions [10,11,12]. Multi-functional bioactive effects have been described for β-casein-derived proteins, such as immunostimulatory, opioid, and angiotensin-converting enzyme (ACE)-inhibitory activities [10,13,14]. There is still little information in the literature on the use of plant proteases to obtain bioactive peptides as products of casein hydrolysis. Moreover, there is a great demand for products that exploit the potential of milk-derived bioactive peptides. These peptides offer a promising approach to preventing, controlling, and even treating lifestyle-related diseases. In vitro determination of biological activities of peptides may be time consuming and expensive [15]. Bioinformatic analysis (in silico) can be an effective tool treated similarly to a preliminary “qualifier” for the evaluation of proteases for the production of peptides with biological activity [16,17,18].

The aim of this study was to determine the proteolytic activity of curly kale leaf extract and to investigate the casein degradation products. In addition, an in silico analysis of the biological activity of the casein-derived peptide sequences was performed.

## 2. Materials and Methods

### 2.1. Plant Material

The curly kale (*Brassica oleracea* L. var. *sabellica* L.) leaves used in this study were harvested from our own cultivation (located in southeastern Poland) in October 2020. The fresh curly kale leaves were washed, dried, and freeze-dried by lyophilization and then stored in a freezer at −20 °C until analysis.

### 2.2. Preparation of Curly Kale Leaf Extract

Lyophilized curly kale leaves (1 g) were mixed with 20 mM phosphate buffer pH 7.0 (15 mL). Then, the cells were disrupted by sonication (Sonics VCX 750 Vibra-Cell™, ultrasonic processor, Newtown, CT, USA) in ice (6 × 30 s, 5 min breaks on ice, 50%). Insoluble particles were removed by centrifugation (10,000 rpm, 20 min, 4 °C), and the supernatant was transferred to a new tube and used for hydrolysis of casein.

### 2.3. Hydrolysis of Casein by Curly Kale Extract

Casein hydrolysates were prepared according to Egito et al. [19], with minor modifications. For casein hydrolysis by curly kale leaf extract, a 5% (*v*/*v*) solution (in water) of casein from bovine milk (Sigma-Aldrich, Saint Louis, MO, USA) was dissolved in 20 mM sodium phosphate buffer (pH = 7.0) to obtain a 0.5% final solution. Then, 2 mL of curly kale leaf extract was added to 2 mL of 0.5% casein solution. Subsequently, 1 mL aliquots of this mixture were transferred to new Eppendorf tubes, and the hydrolysis reaction was carried out at 33 °C in a thermoblock. For further analysis, the samples were collected after 1 h, 4 h, 8 h, and 24 h of incubation. The hydrolysis reaction was inhibited by heating the hydrolysate at 100 °C for 5 min. Samples were stored at −20 °C until further analysis. The casein degradation was visually observed using SDS–PAGE and determined by increasing the concentration of peptide after hydrolysis time.

### 2.4. Sodium Dodecyl Sulfate–Polyacrylamide gel Electrophoresis (SDS–PAGE)

Before electrophoretic separation, the protein concentration of the curly kale extract and hydrolysates was assayed according to the Bradford method [20]. The degree of hydrolysis of casein by curly kale extract was determined using SDS–PAGE [21]. To this end, the sample hydrolysates were mixed with Laemmli buffer (containing 0.3125 M Tris-HCl pH 6.8, 10% (*w*/*v*) SDS, 50% (*v*/*v*) glycerol, 25% (*v*/*v*) 2-mercaptoethanol, 0.02% (*w*/*v*) bromophenol blue) in a 1:4 ratio. Samples were denatured by heating at 100 °C for 5 min. Gel electrophoresis was performed using 12% resolving polyacrylamide gel and 5% stacking gel in Tris-glycine buffer (0.025 M Tris, 0.192 M glycine, 0.1% (*w*/*v*) SDS). Electrophoresis was carried out at a constant voltage of 120 V until the dye front almost reached the bottom of the gel. To visualize proteins, the gel was stained with 0.25% (*w*/*v*) Coomassie brilliant blue R250 (dissolved in a solution of 10% (*v*/*v*) glacial acetic acid, 50% (*v*/*v*) methanol and 40% (*v*/*v*) deionized water) for approximately 4 h. Afterward, the gel was decolorized using a destaining solution (40% (*v*/*v*) methanol, 10% (*v*/*v*) acetic acid and 50% (*v*/*v*) deionized water).

### 2.5. Determination of Peptide Concentration

The concentration of peptides obtained after protein hydrolysis was determined by Direct Detect Quantitation equipment (Direct Detect Spectrometer, Merck Millipore, Burlington, MA, USA). The above method uses the fact of absorbing electromagnetic radiation in multiple regions of the mid-IR spectrum including the strong band at 1600–1690 cm^−1^, which is assigned to C=O stretching vibrations of the peptide bond.

For analysis, proteins in all tested samples were precipitated by 10% trichloroacetic acid. After 15 min of incubation, samples were centrifuged at 5500× *g* for 10 min at 4 °C. Next, 2 µL of obtained supernatants containing peptides were spotted onto an assay-free membrane card and inserted into a spectrometer. BSA standard curve was stored in the instrument.

### 2.6. Mass Spectrometry Analysis

Mass spectrometry analysis was used to test the effectiveness of the enzyme and to identify potential proteolytic cleavage sites. Mass spectrometry analysis was performed for the potential peptides and proteins of each treated sample and, for control purposes, also for plant extract and untreated casein solution from bovine milk.

First, all tested samples—casein, plant extract containing proteolytic enzyme, and hydrolyzed casein samples—were desalted and concentrated using C18 Zip-TIP pipette tips for molecular mass less than 10 kDa and C4 Zip-TIP for molecular mass greater than 10 kDa according to the manufacturer’s instructions (Merck Chemicals, Billerica, MA, USA, PR 02358, Technical Note). Prepared solutions were applied to an AnchorChip plate (Bruker, Bremen, Germany) in two ways. To obtain spectra for peptides of molecular mass below 10 kDa, samples were spotted on the plate and covered with an α-cyano-4-hydroxycinnamic acid matrix (HCCA, Bruker, Bremen, Germany). After drying, spectra were recorded in active positive reflector mode within the 0.7–5 kDa m/z range. For molecular mass greater than 10 kDa, samples were mixed 1:1 (*v*:*v*) with HCCA and DHB (2,5-dihydroxy benzoic acid, Bruker, Bremen, Germany) solution. The mixtures were spotted on an AnchorChip plate previously covered with saturated HCCA in acetone Matrix-assisted laser desorption/ionization–time of flight (MALDI–TOF) mass spectrometric analysis of intact proteins larger than 100 kDa [22]. After drying, spectra were recorded in active positive reflector mode within the 13–30 kDa m/z range.

All mass spectra were recorded using an UltrafleXtreme MALDI–TOF/TOF spectrometer (Bruker, Bremen, Germany) and flexControl 3.3 (Bruker, Bremen, Germany) software. Collected spectra were smoothed and baseline corrected. The peak list generated in flexAnalysis 3.0 software for a signal-to-noise ratio of >3 was transferred to BioTools 3.2 (Bruker, Bremen, Germany). To determine the probable peptide sequences or their parts in enzyme-treated samples, the peptide ions were fragmented using the LIFT mode (detector gain Boost: 100%; Laser Power Boost: 40% and Analog Offset: 0.5%) and subjected to BioTools RapiDeNovo sequencing. Errors used in RapidDeNovo were as follows: Mass Tol. MS: 0.1–0.5 Da; MS/MS Tol.: 0.1–0.5 Da. The method did not indicate specific N-terminal and C-terminal amino acids, and the search was performed using the full pool of amino acids. Designated peptide sequences or their fragments were further analyzed using the Basic Local Alignment Search Tool (BLAST; https://blast.ncbi.nlm.nih.gov/Blast.cgi (accessed on 20 May 2021)) homology database. This database allowed the peptides presented in the hydrolyzed sample to be matched (in the homology field) with analogous peptides found in the casein types.

### 2.7. In Silico Analysis

In silico studies of the obtained peptides were conducted. The bioactivity estimation of the peptides was performed using the BIOPEP Data Base, which provides enzymatic hydrolysis, bioactivity prediction, and comparison of possible achieved peptides in their databases [23]. Milk Bioactive Peptides and Feptide Data Bases were also used [24,25].

## 3. Results and Discussion

### 3.1. Electrophoretic Pattern of Bovine Milk Casein Treated with Curly Kale Extract

Our previous study has revealed that curly kale juice can be an effective milk coagulant in feta-type cheese production [9]. However, no data on the pattern of bovine milk casein hydrolysis by curly kale extract was studied in detail. The hydrolytic changes in casein hydrolyzed by curly kale extract were monitored by SDS–PAGE (Figure 1). Comparison of electrophorograms showed differences in the protein–peptide profile during incubation. As incubation time progressed, the quantity of low-molecular-weight products increased, while both the intensity and quantity of larger compounds (which corresponded to α-, β- and κ-casein) decreased. Thus, after 24 h, all of the bovine caseins had been degraded by the curly kale extract through proteolysis (Figure 1, lane 5). The α- and β-casein fractions appeared to be initially more resistant to curly kale extract hydrolysis than κ-casein, which was absent in the protein profile after 8 h of incubation (Figure 1, lane 4). Similar results were reported for protease extracts from tamarillo (*Solanum betaceum*) fruit, which also resulted in the hydrolysis of caseins into large peptides [26]. After 24 h of incubation, all casein fractions, as well as β-lactoglobulin, were fully hydrolyzed (Figure 1, lane 6). Our results are in line with other studies indicating that plant-derived coagulants are particularly active on the four types of caseins: αs1-, αs2-, β-, and κ-casein [27]. However, the pattern of hydrolysis of α- and β-casein strongly depends on the plant species. In the case of proteases from *Cynanchum otophyllum*, β-casein and κ-casein were completely hydrolyzed after 4 h [28], whereas only partial hydrolysis of α-casein was observed. On the other hand, Anusha et al. [29] found an extract of *Calotropis gigantea* effective in degradation of κ-casein in just one hour.

The casein degradation was confirmed by quantitative analyzes of peptide concentrations in samples after hydrolysis over time. Thus, the peptide concentrations for the samples after 1, 4, 8 and 24 h of incubation were, respectively, 2.57, 3.57, 4.22, and 4.33 mg/mL. The protein degradation can be also easily observed on an SDS–PAGE gel. During the hydrolysis time, caseins bands were less visible at the cost of creating stronger bands for masses below 10 kDa. Based on these results, we can determine the eight-hour time of hydrolysis as sufficient, after which the quantity of peptides no longer increases significantly.

### 3.2. MALDI–TOF MS Analysis

The spectrum shows signals corresponding to the presence of certain peptides in the casein sample (Figure 2). These peptides are not present in the samples after hydrolysis, indicating that they have been hydrolyzed to shorter fragments, including those smaller than 700 Da.

Signals for α-, β-, and κ-caseins are seen around 20 kDa. A series of peaks can be seen in the vicinity of 12 kDa (Figure 3). These may indicate the residues of γ-casein [30], but the SDS–PAGE analysis of the electrophoretic gel with the lack of proteins with such mass reveals that more likely, the occurrence of this signal is correlated with the presence of higher states of charge such as z = 2. For the [M + 2H]^2+^ ions formed during the ionization, the mass value is 0.5 m/z [M + H]^+^. In the case of beta casein, the m/z for the pseudomolecular ion is 23.974 Da, so half the mass is 11.987 Da. The generation of this type of ions is often observed in the case of larger molecules ionization in the MALDI–TOF technique. This process is spontaneous and uncontrolled, especially during the high laser power operations.

On the mass spectra determined for the range of 0.7–5 kDa, characteristic peaks formed as a result of hydrolysis by a preparation containing curly kale extract are clearly visible (Figure 4A and Figure 5A). In the case of spectra determined for typical protein masses, signals for caseins are absent or present with low intensity (Figure 4B and Figure 5B). Peaks were fragmented in a collision chamber in TOF/TOF LIFT mode. Nine of the detected peptide signals were assigned peptide sequences by matching probable amino acid fragments using the RapidDeNovo method (BioTools 3.2, Bruker). The resulting fragments were assigned to casein molecules in the BLAST matching database, and the complete sequence was determined using PeptideMass (Expasy).

The peptides determined were fragments of β-casein, most likely because its content was highest among caseins. In addition, the mass spectra of the hydrolyzed samples for the casein bands show that the β-casein peak is absent or of negligible intensity (Figure 4 and Figure 5).

Based on the sequence analysis, seven bioactive peptides were identified with molecular mass in the range of 1100–3000 Da (Table 1). Thus, we can state that the identified peptides are products of hydrolysis of the initial material by proteolytic enzymes present in the curly kale extract. Taking into account in silico analysis of determined peptide sequences and the amino acids preceding the identified peptides, we can initially indicate that enzyme from curly kale extract cleaves protein at the N-terminal between various residues. There are insufficient data to be able to clearly define the sites of the enzyme cleavage and be sure that the enzyme is acting specifically.

Milk proteins are rich sources of bioactive peptides [6,18]. In this study, we determined the in silico potential of curly kale extract from generated peptides from casein. The use of plant proteases in the formation of bioactive peptides is still scarce. The cysteine proteases papain and bromelain have been the plant proteases most studied for this purpose. Only three vegetables from the *Brassicaceae* family have been reported as sources of proteases, which actively generate bioactive peptides. These include broccoli, canola, and rapeseed. All of them exhibited antihypertensive activity, which was confirmed in in vitro study [6]. Depending on their composition and sequence, bioactive peptides may exhibit a number of different activities in vivo, which affect, e.g., the cardiovascular, digestive, immune, and nervous systems. In our study, all detected peptides were products of β-casein hydrolysis using curly kale leaf extract, and in silico analysis using three protein databases [23,24,25] revealed their possible multi-functional bioactive effects. All seven identified peptides can possibly show angiotensin-I-converting enzyme (ACE) inhibitory activity, dipeptidyl peptidase IV inhibitory activity, and a glucose uptake stimulation effect. These activities have been shown to counteract metabolic syndrome (high blood pressure, high blood sugar, unhealthy cholesterol levels, and abdominal fat), which is associated with cardiovascular disease and type 2 diabetes. Other studies have also confirmed the antihyperglycemic and antihypertensive effects of peptides resulting from casein hydrolysis [14,31]. High antihypertensive activity has been described for tripeptides derived from β-casein following fermentation of milk by *Lactobacillus helveticus* and *Saccharomyces cerevisiae* [32]. Other important bioactivities that possibly can be attributed to peptides obtained after hydrolysis of casein by curly kale proteases include antioxidant, immunomodulatory, and cytomodulatory effects. All of these mechanisms are involved in the body’s response to inflammation, which initiates many human chronic diseases. Li et al. [33] obtained five novel peptides with high antioxidant activity from goat milk casein hydrolyzed by a combination of neutral and alkaline protease. Only a few studies have reported proteolytic activity of extracts prepared from plants. Corrons et al. [34] showed that proteolytic enzymes from *Maclura pomifera* are able to release ACE inhibitory peptides from bovine caseins. Casein hydrolysate obtained with aspartic proteinases present in extracts from artichoke flower (*Cynara scolymus* L.) was shown to have higher antioxidant, antimicrobial, and angiotensin-I-converting enzyme (ACE) inhibitory activity, compared with the findings of similar studies using enzymes of animal or bacterial origin [35].

## 4. Conclusions

This short report, although preliminary, may have potentially high scientific value, as it was the first to determine the capacity for bioactive peptide production as a result of β-casein hydrolysis by proteases from curly kale leaf extract. Taking into account in silico analysis of determined peptide sequences and the amino acids preceding the identified peptides, we can initially indicate that enzyme from curly kale extract cleaves proteins at the carboxyl side of valine, serine, phenylalanine, threonine, or leucine. We can assume that enzyme mainly and preferentially cleaves peptide bond after nonpolar amino acids. Nevertheless, at the moment, there are insufficient data to be able to clearly define the sides of the enzyme cleavage and be sure that the enzyme is not acting nonspecifically. As indicated by in silico analysis, the β-casein-derived peptides detected could potentially display ACE inhibitory, antioxidant, dipeptidyl peptidase IV inhibitory, antithrombotic, immunomodulatory, and antiamnesic activity. However, further studies are needed to elucidate their potential physiological effects on the human body. In subsequent studies, they will be chemically synthesized for validation of relevant activities in animal models and in cell line cultures. The results of this work also provide a basis for further applications of curly kale proteases in the food industry. They may be especially useful in the development of functional foods and nutraceuticals.

## Figures and Tables

**Figure 1 foods-10-02877-f001:**
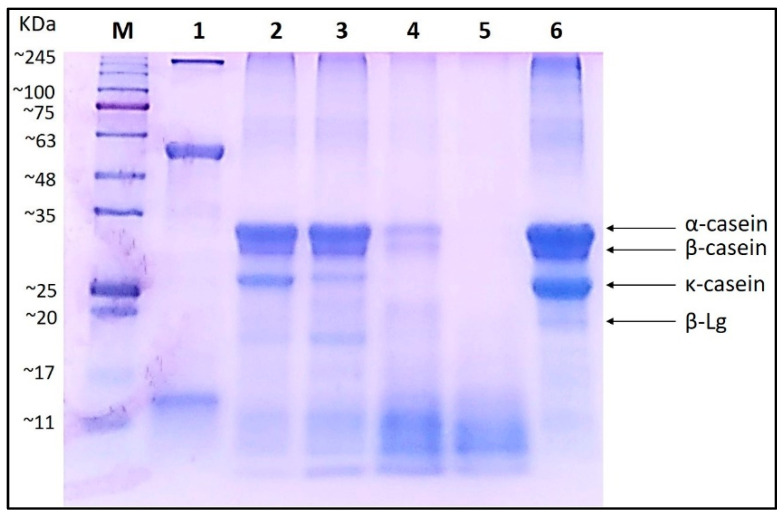
Electrophoresis of bovine milk casein hydrolyzed by curly kale leaf extract. M—prestained protein molecular weight marker; lane 1—curly kale extract; lanes 2 to 5—casein hydrolyzed by curly kale extract after 1 h, 4 h, 8 h, and 24 h, respectively; lane 6—solution of bovine milk casein (10 μL of 0.25%).

**Figure 2 foods-10-02877-f002:**
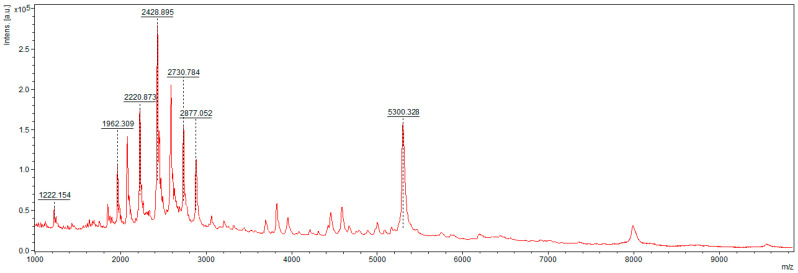
Spectrum obtained for casein at low molecular masses (1–10 kDa) using HCCA matrix.

**Figure 3 foods-10-02877-f003:**
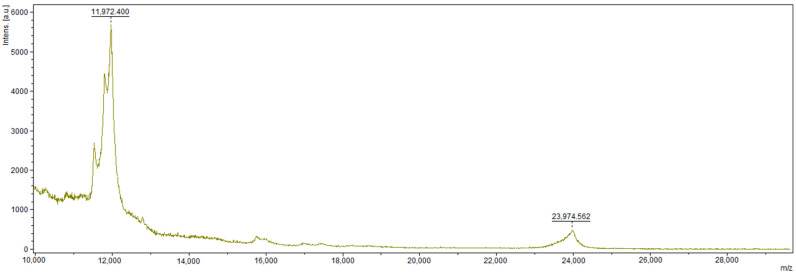
Spectrum obtained for casein in the 10–30 kDa molecular mass range using HCCA+DHB matrix (HCCA in acetone).

**Figure 4 foods-10-02877-f004:**
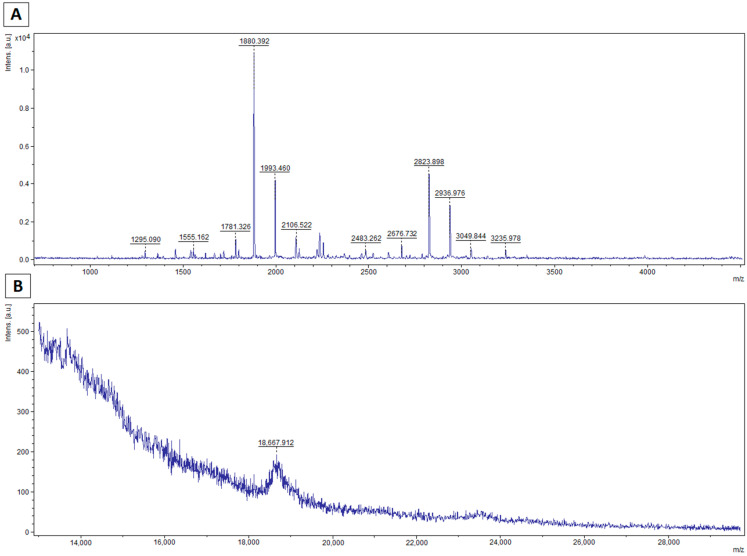
Spectra acquired for casein treated with curly kale extract for 8 h (**A**) for low molecular masses (0.7–5 kDa) using the HCCA matrix; (**B**) for molecular masses in the casein range (13–30 kDa) using the HCCA+DHB matrix.

**Figure 5 foods-10-02877-f005:**
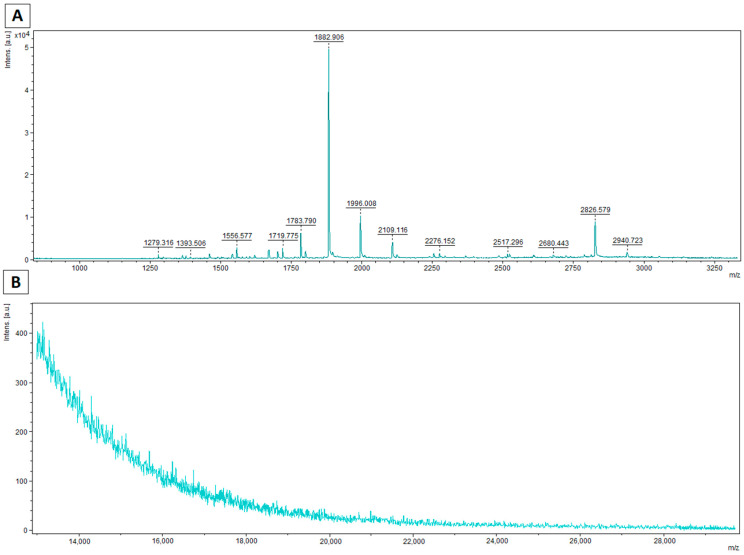
Spectra acquired for casein treated with curly kale extract for 24 h (**A**) for low molecular masses (0.7–5 kDa) using the HCCA matrix; (**B**) for molecular masses in the casein range (13–30 kDa) using the HCCA+DHB matrix.

**Table 1 foods-10-02877-t001:** Assigned peptide sequences by RapidDeNovo method.

Mass (Da)	Peptide Sequence	Source of Peptide	ID of the Bioactive Peptide in the Database	Activity
1151	LGPVRGPFPIIV	β-casein	(3169, 8594, 8882, 8857, 8801, 8506) ^c^	dipeptidyl peptidase IV inhibitor
			(7512, 7619, 8160, 7508, 3502, 7545) ^c^	ACE inhibitor
			8286 ^c^	antioxidative
			2753 ^c^	regulating the stomach mucosal membrane activity
			3283 ^c^	antithrombotic
			3461 ^c^	antiamnesic
			8325 ^c^	glucose uptake stimulating
1459	PGPIPNSLPQNIPPLT	β-casein	(3169, 8824, 3180, 8875, 3170, 8638) ^c^	dipeptidyl peptidase IV inhibitor
			(7512, 7836, 7625, 7513)^c^	ACE inhibitor
			(2753, 2756, 2754)^c^	regulating the stomach mucosal membrane activity
			(3283, 3284, 3285) ^c^	antithrombotic
			(3461, 3459, 3460) ^c^	antiamnesic
1718	YQEPVLGPVRGPFPIIV	β-casein	biopep01621 ^a^	antihypertensive
			biopep04091 ^a^; P02666 ^b^	antimicrobial
			biopep04801 ^a^; P02666 ^b^	immunomodulatory and cytomodulatory
			(3169, 8869, 8922, 8594, 8593, 8882) ^c^	dipeptidyl peptidase IV inhibitor
			P02666 ^b^(7512, 7619, 8160, 7508, 3502, 7545) ^c^	ACE inhibitor
			(8286, 7878, 7879) ^c^	antioxidative
			2753 ^c^	regulating the stomach mucosal membrane activity
			P02666 ^b^; 3283 ^c^	antithrombotic
			3461 ^c^	antiamnesic
			8325 ^c^	glucose uptake stimulating
1881	LYQEPVLGPVRGPFPIIV	β-casein	P02666 ^b^	immunomodulatory
			(3169, 8869, 8922, 8594, 8593 8857) ^c^	dipeptidyl peptidase IV inhibitor
			(7512, 7619, 3381, 8160, 7508, 3502) ^c^	ACE inhibitor
			(7872, 7879, 7878, 8286) ^c^	antioxidative
			2753 ^c^	regulating the stomach mucosal membrane activity
			8325 ^c^	glucose uptake stimulating
			3283 ^c^	antithrombotic
			3461 ^c^	antiamnesic
2281	WMHQPHQPLPPTVMFPPQSVLS	β-casein	(3180, 8923, 8877, 8532, 8856, 8638) ^c^	dipeptidyl peptidase IV inhibitor
			(3385, 7836, 3502, 3391, 2664, 7513) ^c^	ACE inhibitor
2824	LSQSKVLPVPQKAVPYPQRDMPIQAFL	β-casein	(3180, 8877, 8922, 3181, 3171, 8857) ^c^	dipeptidyl peptidase IV inhibitor
			(3373, 2653, 3370, 8951, 7583, 3666) ^c^	ACE inhibitor
			(8479, 7877, 7896, 7875, 7876, 8278) ^c^	antioxidative
3024	LSLSQSKVLPVPQKAVPYPQRDMPIQAFLL	β-casein	(3182, 3180, 8877, 8867, 8922, 3181) ^c^	dipeptidyl peptidase IV inhibitor
			(3373, 2653, 3370, 8951, 7583, 3666) ^c^	ACE inhibitor
			(8479, 7877, 7896, 7875, 7876, 8278) ^c^	antioxidative

^a^ BIOPEP DB [23],^b^ MBP DB [24] ^c^ FeptideDB [25].

## Data Availability

The datasets generated for this study are available on request to the corresponding author.

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
