# Peer review of "Potential Biological Activities of Peptides Generated during Casein Proteolysis by Curly Kale (Brassica oleracea L. var. sabellica L.) Leaf Extract: An In Silico Preliminary Study"

_foods, 2021, doi:10.3390/foods10112877_

Round 1

Reviewer 1 Report

This brief report is an original scientific report that, in my opinion, fits perfectly into the aims and scope of the scientific journal Foods. Because this is an unprecedented scientific report on the possibility of applying proteases from curly kale leaf extract to generate bioactive peptides from the hydrolysis of Beta Casein in milk. It is also noteworthy that the work now submitted for evaluation by this renowned scientific journal represents the continuation of an article recently published (2020) by the authors in which they highlight the potential use of the same extract for the production of Feta cheese.

The introduction is enough to support the reader as to the importance of carrying out the study presented. In this, the objective of the study was presented, at the end of this topic, in a clear and direct way, being consistent with the methodology used in the study. Furthermore, in the introduction, adequate and mostly up-to-date references were used.

As presented in the text, the research project that resulted in this scientific report presents an appropriate, sequential and very well described methodology, which enables its faithful reproduction by those interested.

The results were presented in a logical sequence that attracts and keeps the reader involved in the continuity of the reading. The text presents fluidity and values ​​the discussion of results and comparison of these with those already published by other authors. However, it fails when trying to explain the site of protease hydrolysis from curly leaf extract on the protein polypeptide chain (lines 224-229). The explanation given, despite being speculative, seems not very clear and convincing. Which leads us to question: If the enzyme hydrolyzes the polypeptide chain at the N-terminal side containing amino acids such as valine, serine, phenylalanine and leucine (cleaving the peptide bond after nonpolar amino acid residues), why in the seven identified bioactive peptides (table 1), only leucine appears as an N-terminal amino acid in the sequence of four of the seven identified peptides ? In the other 3 peptides, proline, tyrosine and tryptophan are found as N-terminal residues instead of valine, serine and phenylalanine.The way this paragraph was written, it generates the gap in understanding that leads me to the question above. Faced with this doubt, in order to improve the quality of the submitted manuscript, I suggest that the Editorial ask the authors to answer this question, conditioning the possibility of accepting the work to a more plausible explanation of the content presented in this paragraph. That along with the answer, there could already be a new wording to the paragraph in response to the question made. In the following paragraph (lines 230-231) where it is written: "In this study, we determined the potential of curly kale extract of the generate peptides from casein in silico", I suggest that it be written as follows: In this study, we determined the in silico potential of curly kale extract from generate peptides from casein." I also suggest that in line 236, "in vitro" is written in italics. In the other sentences of the results and discussions, the texts are well written and discussed, reinforcing the importance and innovative results obtained by in silico analysis of peptide sequences obtained by mass spectrometry of casein hydrolysates.

Regarding the conclusion, the first written sentence (lines 263-265) seems to need a complement to understand its meaning. In general, as the conclusion is written, it does not express in a summarized form the important results obtained, especially with regard to the proteolytic activity of curly kale leaf extracts, its supposed site of hydrolysis and the degradation products of caseins . Despite this, it highlights the potential of in silico analysis, through the possible activities identified based on the sequences of the seven bioactive peptides identified, and also describes the directions that will possibly be taken for the continuation of the work. Thus, I recommend that a new essay be given the conclusion in the final document.

The references were all checked as cited in the text and are appropriate to the research area considered in the scientific report.

With respect to errors in the text, they are minimal and quick to correct. Listed below are the observed:

- Line 27: "possibility" and not "possibility
- Line 222 (Table 1): "activity" or "activ-ty" and not "activ-ity"
- Line 222 (Table 1): "stimulating" or "stimila-ting" and not "stimulat-ing"
- Line 227: "phenylalanine" and not "phenyloalanine"
- Line 236: "in vitro" (italic) and not "in vitro" (normal).
- Line 275: "preparation" and not "prepa-ration"
- Line 275-276: "visualization" or "vi-sualization" and not "vis-ualization"
- In the references of lines 287, 292 and 294, put the year in bold.

Author Response

Reviewer 1

1. The results were presented in a logical sequence that attracts and keeps the reader involved in the continuity of the reading. The text presents fluidity and values ​​the discussion of results and comparison of these with those already published by other authors. However, it fails when trying to explain the site of protease hydrolysis from curly leaf extract on the protein polypeptide chain (lines 224-229). The explanation given, despite being speculative, seems not very clear and convincing. Which leads us to question: If the enzyme hydrolyzes the polypeptide chain at the N-terminal side containing amino acids such as valine, serine, phenylalanine and leucine (cleaving the peptide bond after nonpolar amino acid residues), why in the seven identified bioactive peptides (table 1), only leucine appears as an N-terminal amino acid in the sequence of four of the seven identified peptides ? In the other 3 peptides, proline, tyrosine and tryptophan are found as N-terminal residues instead of valine, serine and phenylalanine.The way this paragraph was written, it generates the gap in understanding that leads me to the question above. Faced with this doubt, in order to improve the quality of the submitted manuscript, I suggest that the Editorial ask the authors to answer this question, conditioning the possibility of accepting the work to a more plausible explanation of the content presented in this paragraph. That along with the answer, there could already be a new wording to the paragraph in response to the question made.

This is a very valid point. The paper contains an incorrectly worded assumption, which has been corrected. Taking into account in silico analysis of determined peptide sequences and the amino acids preceding the identified peptides we can initially indicate that enzyme from curly kale extract cleaves protein at the N-terminal residues as is pointed below:

MW [Da]

Sequence

Source

1151

L.GPVRGPFPIIV-

β-casein

1459

P.GPIPNSLPQNIPPL.T

β-casein

1718

Y.QEPVLGPVRGPFPIIV-

β-casein

1881

L.YQEPVLGPVRGPFPIIV-

β-casein

2281

W.MHQPHQPLPPTVMFPPQSVL.S

β-casein

2824

L.SQSKVLPVPQKAVPYPQRDMPIQAF.L

β-casein

2937

L.SQSKVLPVPQKAVPYPQRDMPIQAFL.L

β-casein

3024

L.SLSQSKVLPVPQKAVPYPQRDMPIQAFL.L

β-casein

>AAA30431.1 beta-casein [Bos taurus]MKVLILACLVALALARELEELNVPGEIVESLSSSEESITRINKKIEKFQSEEQQQTEDELQDKIHPFAQTQSLVYPF|PGPIHNSLPQNIPPLT|QTPVVVPPFLQPEVMGVSKVKEAMAPKHKEMPFPKYPVEPFTESQSLTLTDVENLHLPLPLLQS|WMHQPHQPLPPTVMFPPQSV|LS|LSQSKVLPVPQKAVPYPQRDMPIQAFL|L|YQEPV|LGPVRGPFPIIV

F /P        phenylalanine/ proline                 side chain nonpolar / nonpolar

T/Q        treonine/glutamine                    polar/polar

S/W       serine / tryptophan                        polar / nonpolar

V/L         valine / leucine                              nonpolar /nonpolar

S/L         serine / leucine                              polar / nonpolar

L/L          leucine / leucine                              nonpolar / nonpolar

L/Y         leucine / tyrozine                            nonpolar / polar

V/L         valine / leucine                               nonpolar / nonpolar

We can admit that based on obtained data we can not state that enzyme preferentially cleaves peptide bond after nonpolar amino acids. Nevertheless, at the moment there is insufficient data to be able to clearly define the sides of the enzyme cleavage and be sure that the enzyme is acting specifically. The main aim of our work was to prove that the enzyme has a proteolytic activity, resulting in obtaining a number of peptides not present in the initial material. In our study, for the determination of the peptide composition as well as for the preliminary analysis of potential cleavage sites, we selected only high-score results, which are very reliable.

Appropriate changes has been included in the text according to Reviewers suggestion.

2. In the following paragraph (lines 230-231) where it is written: "In this study, we determined the potential of curly kale extract of the generate peptides from casein in silico", I suggest that it be written as follows: In this study, we determined the in silico potential of curly kale extract from generate peptides from casein." - has been corrected according to the reviewer's suggestion

3. I also suggest that in line 236, "in vitro" is written in italics. In the other sentences of the results and discussions, the texts are well written and discussed, reinforcing the importance and innovative results obtained by in silico analysis of peptide sequences obtained by mass spectrometry of casein hydrolysates. - has been corrected

4. Regarding the conclusion, the first written sentence (lines 263-265) seems to need a complement to understand its meaning. In general, as the conclusion is written, it does not express in a summarized form the important results obtained, especially with regard to the proteolytic activity of curly kale leaf extracts, its supposed site of hydrolysis and the degradation products of caseins . Despite this, it highlights the potential of in silico analysis, through the possible activities identified based on the sequences of the seven bioactive peptides identified, and also describes the directions that will possibly be taken for the continuation of the work. Thus, I recommend that a new essay be given the conclusion in the final document. - Conclusion has beer rewritten according to the reviewer's suggestion:

„ This short report, although preliminary may have potentially high scientific value, as it was the first to determine the capacity for bioactive peptide production as a result of β-casein hydrolysis by proteases from curly kale leaf extract. Taking into account in silico analysis of determined peptide sequences and the amino acids preceding the identified peptides we can initially indicate that enzyme from curly kale extract cleaves proteins at the carboxyl side of valine, serine, phenylalanine, threonine or leucine. We can assume that enzyme mainly and preferentially cleaves peptide bond after nonpolar amino acids. Nevertheless, at the moment there is insufficient data to be able to clearly define the sides of the enzyme cleavage and be sure that the enzyme is not acting nonspecifically. As indicated by in silico analysis the β-casein-derived peptides detected could potentially display ACE inhibitory, antioxidant, dipeptidyl peptidase IV inhibitory, antithrombotic, immunomodulatory, and antiamnesic activity. However, further studies are needed to elucidate their potential physiological effects on the human body. In subsequent studies, they will be chemically synthesized for validation of relevant activities in animal models and in cell line cultures. The results of the present work also provide a basis for further applications of curly kale proteases in the food industry. They may be especially useful in the development of functional foods and nutraceuticals.”

5. With respect to errors in the text:

- Line 27: "possibility" and not "possibility  - has been corrected

- Line 222 (Table 1): "activity" or "activ-ty" and not "activ-ity"- has been corrected

- Line 222 (Table 1): "stimulating" or "stimila-ting" and not "stimulat-ing"- has been corrected

- Line 227: "phenylalanine" and not "phenyloalanine"- has been corrected

- Line 236: "in vitro" (italic) and not "in vitro" (normal). - has been corrected

- Line 275: "preparation" and not "prepa-ration"- has been corrected

- Line 275-276: "visualization" or "vi-sualization" and not "vis-ualization"- has been corrected

- In the references of lines 287, 292 and 294, put the year in bold. - has been corrected

Reviewer 2 Report

This paper is focused on the evaluation of the proteolytic activity of curly kale leaf extract against casein.

After a thorough analysis of this work it is the opinion of this referee that it has to be reviewed in depth. In general terms this work is not well written and it is not easy to read. English language is sometimes quite poor and should be overall revised.

Below are some tips to consider

Lines 19, 93, 97 and others: the authors state that the degree of casein hydrolysis was determined using SDS-PAGE but they do not provide a description of how this technique was used for a quantitative assessment of the degree of hydrolysis

Lines 196, 197: "Signals for α-, β-, and κ-caseins are seen around 20 kDa. A series of peaks can be seen in the vicinity of 12 kDa (Figure 3). These may indicate the presence of γ-casein". However, there is no evidence of γ-casein in the SDS page of Figure 1

Lines 205/206: Authors do not explain on what basis nine of the detected peptide signals were selected

There are also errors in the text. E.g.

Line 27: possibi-lity

Line 49: previous study have

Line 227: phenyloalanine

Author Response

Reviewer 2

  1. Lines 19, 93, 97 and others: the authors state that the degree of casein hydrolysis was determined using SDS-PAGE but they do not provide a description of how this technique was used for a quantitative assessment of the degree of hydrolysis

The authors are grateful to the reviewer for pointing out inaccuracies. The method indicated was for a preliminary comparison of protein-peptide profiles and to assess changes (visible in polyacrylamide gel) products obtained after hydrolysis with curly kale extract.  Determination of the level of hydrolysis degree in tested as well as factors influencing the process will be the subject of further research. Therefore, to clarify the presented information appropriate changes were made in the text (line 19-20): ”The changes of patterns in protein and peptides profiles of products after hydrolysis by curly kale extract was determined using SDS-PAGE.”

We changed the expression "degree of hydrolysis" in the text to a more appropriate "hydrolysis efficiency". The casein degradation was confirmed by quantitative analyzes of peptide concentrations in samples after hydrolysis over time. The protein proteolysis can be also easily observe on SDS-PAGE gel. During the hydrolysis time, caseins bands were less visible at the cost of creating stronger bands for masses below 10 kDa.

  1. Lines 196, 197: "Signals for α-, β-, and κ-caseins are seen around 20 kDa. A series of peaks can be seen in the vicinity of 12 kDa (Figure 3). These may indicate the presence of γ-casein". However, there is no evidence of γ-casein in the SDS page of Figure 1

After a thorough analysis of the reported results and comparing them again with the results indicated in the references [30], we noticed that instead of gamma it should be para-κ-casein. However, too many products of similar mass (in the range of 17-10 kDa) are visible in the electrophoretic image (in Figure 1) as the regions of smear in the separation lines of the products after hydrolysis (lanes 2 to 5 - casein hydrolyzed by curly kale extract after 1 h, 4 h, 8 h, and 24 h) not allow to unequivocally indicate specific bands of para- κ-caseins. Therefore the text (lines 204-205) was corrected as follows: “Signals for α-, β-, and κ-caseins are seen around 20 kDa. A series of peaks can be seen in the vicinity of 12 kDa (Figure 3) indicating the presence of para-κ-casein that corresponds with findings described by [30].”

After consultation with manufacturer of the MALDI spectrometer we conclude that the main reason of 12 kDa peak occurrence is the presence of higher states of charge, such as z = 2. For the [M + 2H]2+ ions formed during the ionization mass value is 0.5 m/z [M + H] +. In the case of beta casein, the m/z for the pseudomolecular ion is 23.974 Da, so half the mass is 11.987 Da. The generation of this type of ions is often observed in the case of ionization of larger molecules in the MALDI technique. This process is spontaneous and uncontrolled, especially during the high laser power ionization. The above information was included in the manuscript.

  1. Lines 205/206: Authors do not explain on what basis nine of the detected peptide signals were selected

We tried to identify as many peptides as possible, but taking into account the complexity of the mixture (several proteins) and the lack of information about cleavage sites, we decided to include only peptides whose sequence was assigned by BioTools Bruker software with statistically significant score. We also wanted to sequence the peptide with the highest intensity in the spectrum (1881 m/z).

  1. There are also errors in the text. g.

Line 27: possibi-lity - has been corrected

Line 49: previous study have - has been corrected „previous study has”

Line 227: phenyloalanine - has been corrected

Reviewer 3 Report

In this study, the authors investigated the proteolytic activity of the extract from curly kale leaf by degrading the casein. The κ-casein is hydrolyzed first by the extract, followed by α- and β-casein. Seven peptides were found in casein hydrolysate by MALDI-TOF MS analysis. Furthermore, bioinformatics analysis was performed to reveal the potential bioactivity of seven peptides.

This paper represents interesting results; but has some shortcomings in regard to some data and text. Below I have provided numerous remarks and made additional suggestions for more in-depth presentation of the data;

Given these shortcomings the manuscript requires revision.

Specific Comments:

Page 1, line 20: “potential proteolytic cleavage sites”. In the article, I did not find the description of the potential digestion site of extract from curly kale leaf.

Page 1, line 27-28: This sentence should be placed at the end of the abstract.

Page 2, line 53: Safety evaluation or prediction results of the extract from curly kale leaf should be added here.

In the method section, databases search and silico analysis for identified peptides should be written in detail.

The extract hydrolyzed κ-casein first, but no peptide was derived from κ-casein. Please discuss this fully.

Finally, it is expected that these 7 peptides can be chemically synthesized for validation of relevant activities in the next study.

Author Response

Reviewer 3

  1. Page 1, line 20: “potential proteolytic cleavage sites”. In the article, I did not find the description of the potential digestion site of extract from curly kale leaf. - has been corrected: „The pattern of the enzymatic activity was determined by MALDI-TOF MS analysis.”
  2. Page 1, line 27-28: This sentence should be placed at the end of the abstract. - has been corrected according to the reviewer's suggestion
  3. Page 2, line 53: Safety evaluation or prediction results of the extract from curly kale leaf should be added here.

As suggested by the reviewer, the text has been added: “The use of kale extract as a coagulating agent in the food industry is safe, and due to its well-studied and described health-promoting properties, it increases the functional value of the product [9].

  1. In the method section, databases search and silico analysis for identified peptides should be written in detail. - relevant part in Material and Methods has been added:

2.7 In silico analysis

In silico studies of the obtained peptides were conducted. The bioactivity estimation of the peptides was performed using the BIOPEP Data Base, which provides enzymatic hydroly-sis, bioactivity prediction and comparison of possible achieved peptides in their databases [23]. Milk Bioactive Peptides and Feptide Data Bases were also used [24, 25].

  1. The extract hydrolyzed κ-casein first, but no peptide was derived from κ-casein. Please discuss this fully.

This is perfectly correct. The MALDI-TOF spectra show numerous signals corresponding to the presence of peptides derived from milk proteins (caseins and beta lactoglobulin) for which no sequence has been assigned. It should also be remembered that masses below 700 m/z are not observed on the mass spectrum, so ion masses corresponding to the presence of di- and tripeptides are not observed for us. Considering the decomposition of casein forms, we conclude that the peptides from each casein form are present, but they have not been assigned with high score to the κ form.

In the presented initial research, we analyzed the selected range of products (obtained after hydrolysis) this information is indicated in Lines 141-142: “…..spectra were recorded in active positive reflector mode within the 13–30 kDa m/z range.”

The result of sequence analysis indicated the presence of seven bioactive peptides with molecular mass in the range of 1,100–3,000 Da (Table 1). (The information in the text of the manuscript- lines 229-230). Our findings correspond to results indicated by[28] that was expressed in line 183-185: ”….. the pattern of hydrolysis of α- and β-casein strongly depends on the plant species. In the case of proteases from Cynanchum otophyllum, β-casein and κ-casein were completely hydrolyzed after 4 h [28], whereas only partial hydrolysis of α-casein was observed.”

The authors of the above-mentioned research paper identified the peptide sequences derived from κ-casein. Most of the peptides exhibited molecular weight below 3,000 Da (only one sequence with higher molecular mass was detected).   Presumably, the applied analysis mass range used in the determination method precluded the identification of lower mass peptides in our investigation.

  1. Finally, it is expected that these 7 peptides can be chemically synthesized for validation of relevant activities in the next study. Further research will be undertaken as a follow-up to the studies described in this article.

Round 2

Reviewer 2 Report

The revised paper can be accepted for publication